# Environment, taxonomy, and socioeconomics predict non-imperilment in freshwater fishes

Christina A. Murphy [1,2,8] ✉, J. Andres Olivos [3,8], Ivan Arismendi [4], Emili García-Berthou [5], Sherri L. Johnson[6] & Jason Dunham[7]

Freshwater fishes are among the most threatened taxa, yet conservation assessments remain incomplete for many species. Freshwater fishes provide essential ecosystem services such as food security, recreational opportunities, and cultural significance. Despite heavy alterations to freshwater ecosystems, the reasons for species' sensitivity and resistance to imperilment are unclear. To address this need, we develop a machine learning framework to predict global imperilment status for 10,631 freshwater fish species using a comprehensive set of environmental, socioeconomic, and intrinsic species-level predictors. Using updated IUCN Red List data, we train and validate Random Forest classifiers to distinguish imperiled (Vulnerable, Endangered, Critically Endangered) from non-imperiled species. We examine the relative influence of 52 variables derived from 12 global sources describing extrinsic environmental and socioeconomic factors and intrinsic species-specific characteristics. Our models achieve higher accuracy for non-imperiled species (90.1%) compared to imperiled species (81.8%), reflecting the greater heterogeneity of threats and conditions driving imperilment. Across models, key predictors include habitat variables, taxonomic order, hydrological characteristics, and disturbance indicators, underscoring the interplay between ecology, geography, and human pressures. This integrative, reproducible approach demonstrates the utility of machine learning for guiding proactive conservation and provides a scalable framework for global biodiversity risk assessment.

Biodiversity conservation is often focused on taxa that are most at-risk[1,2]. These species have often declined to levels that have reduced their capacity to recover, and thus require conservation actions that are difficult and very expensive to implement[3,4]. Although there is general agreement that at-risk species deserve immediate attention, conservation activities can also be effectively focused on species that have yet to be listed as imperiled. Addressing non-imperiled species more proactively can offer management solutions that involve fewer regulatory and logistical constraints[2,5]. In practice, both approaches are complementary, and thus it is important not only to recognize species at risk, but also to identify intrinsic and extrinsic characteristics that make species less vulnerable to extinction.

[1]U.S. Geological Survey, Maine Cooperative Fish and Wildlife Research Unit, Orono, Maine, USA. [2]Department of Wildlife, Fisheries, and Conservation Biology, University of Maine, Orono, Maine, USA. [3]Department of Forest Engineering, Resources, and Management, Oregon State University, Corvallis, Oregon, USA. [4]Department of Fisheries, Wildlife, and Conservation Sciences, Oregon State University, Corvallis, Oregon, USA. [5]GRECO, Institute of Aquatic Ecology, University of Girona, Girona, Catalonia, Spain. [6]Pacific Northwest Research Station, USDA Forest Service, Corvallis, Oregon, USA. [7]U.S. Geological Survey, Forest and Rangeland Ecosystem Science Center, Corvallis, Oregon, USA. [8]These authors contributed equally: Christina A. Murphy, J. Andres Olivos. ✉ e-mail: christina.murphy@maine.edu

Understanding a species' status with respect to extinction risk is a necessary, yet insufficient step toward identifying appropriate conservation interventions. Species status assessments typically focus on geographic ranges, population trends, species traits, and threats such as loss of habitat, overexploitation, and impacts of invasive species[6–9]. In most cases, the drivers of imperiled status can be traced back to human influences[10,11]. Species status assessments that collectively address socioeconomic, environmental, and intrinsic (e.g., species traits, taxonomic) factors can provide a more comprehensive evaluation of drivers, improved status predictions, and a clearer understanding of effective solutions for addressing threats to biodiversity[10].

The global biodiversity crisis increasingly affects freshwater fishes, with a rapidly expanding list of species at risk of extinction from a complex and interacting array of threats[12,13]. Widespread habitat degradation and loss are among the most important threats. For example, approximately two-thirds of the world's largest rivers have already been impounded[14], and many more dams are being planned globally[15], posing challenges for species movement, ecosystem functioning, and habitat connectivity[16]. In addition, habitat degradation and loss and risks posed by biological invasions[17,18] associated with competition[19], predation[20,21], disease[22], and hybridization[23] represent major threats to freshwater fishes. These stressors further interact with climate change to threaten species globally[24]. Consequently, nearly one-third of freshwater fishes currently face the threat of extinction[25].

As the most imperiled and diverse vertebrate group globally[12], freshwater fishes present a critical case for expanding conservation tools. We examine whether complementary data, currently absent from formal status assessments, could reveal broader patterns of imperilment across 10,631 species worldwide. A range of globally consistent indicators of environmental, socioeconomic, and intrinsic species-level factors, excluding variables directly used as listing criteria (e.g., population size and trend, range area), are analyzed as predictors of conservation status (Table 1). We also examine the degree to which these factors are able to accurately predict distributions of fish species believed to be at risk or species not recognized as such (non-imperiled). The current conservation status of freshwater fishes used in this study is based on the International Union for Conservation of Nature (IUCN) Red List designations[26].

In this work, we apply machine-learning classification models to predict the global conservation status of freshwater fishes, integrating 52 variables from 12 international sources, and quantify the relative contributions of extrinsic environmental and socioeconomic factors and intrinsic species-specific characteristics (Table 1). We correct for geographic range-size bias by area-weighting correlated ($R^2 > 0.3$) spatial variables (e.g., count of dams, human population; see "Methods"). We initially tuned an ordinal forest model to predict individual IUCN Red List categories, but high rates of misclassification between adjacent categories prompted a shift to a conventional random forest framework, reframing the analysis as a binary classification of species as either imperiled or non-imperiled. Species classified by the IUCN as Near Threatened, or of Least Concern, were considered non-imperiled, whereas species classified as Vulnerable, Endangered, and Critically Endangered were considered imperiled (Fig. 1). Our findings underscore the critical role of intact habitats and fish taxonomy in predicting non-imperilment across species.

## Results and discussion
Model results indicated that extrinsic environment and socioeconomics were most strongly associated with conservation status (Fig. 2), whereas intrinsic species-level factors contributed less than 10% to model assignment. Overall, non-imperiled species tended to occur in regions with greater water availability, moderate impoundment density, minimal habitat alteration, low human footprint, stable gross domestic product, and relatively few habitat types per unit area (Figs. 3,4). Generally, extreme values for environmental and socioeconomic factors were

associated with imperiled species. This was observed even for species predominantly associated with forested lands. These findings are consistent with previous studies suggesting an association between species imperilment and habitat specificity[27,28]. In many cases, relationships of predictors with conservation status were non-linear (Figs. 3, 4).

**Classification by status**
The model predicted conservation status with high accuracy (balanced classification accuracy: 88%), yet performance was lower for imperiled species (81%) than for non-imperiled species (90%), as indicated by misclassification rates in the test dataset (Fig. 2b). This mismatch suggests that conditions associated with non-imperilment are more consistent and better represented than those driving imperilment. This may be unsurprising if fish imperilment is partly influenced by specific or narrow habitat requirements, consumptive values to humans, or other unique traits contributing to their conservation status. Influential predictors of imperilment may thus provide less consistent targets for conservation than predictors associated with non-imperilment. Further, the IUCN represents a global species status, whereas within the extent of a species range, more local threats could be important (e.g.[29]), and not detected with an analysis focused on whole species' ranges. Our analysis is a first step towards systematically evaluating influences and general drivers of imperilment on a global extent. It integrates knowledge of species distributions, other data to inform listings, and predictors to better understand global patterns of imperilment.

The strongest and most unique predictor of the binary random forest model was hydro-geomorphic diversity, where more habitat types per unit area in the species range showed a positive non-linear dependence with imperilment probability (Fig. 2). Higher environmental heterogeneity (low homogeneity) may proxy compromised connectivity among habitat patches, a described driver of population decline in stream fishes[30]. Different measures of water availability, such as stream power and permanent water cover, were among the most influential factors, consistent with their obvious influence on habitat and its relationships with other predictors (Fig. 3). Taxonomic order was the second most influential variable, likely due the fact that fishes within the same order share many traits in common and thus may respond more similarly to environmental stressors (Fig. 2). Taxonomic influence has been reported in previous analyses of the conservation status of small-bodied freshwater fishes, though only hypothesized to occur throughout IUCN Red List designations[31]. Future efforts could determine whether these results are due to common stressors or unavoidable subjectivities inherent in conservation assessment frameworks, despite rigorous implementation[32].

Whereas intrinsic factors had the least cumulative influence on predictions (predominantly in the form of taxonomy), the level of human knowledge on species traits showed high predictive importance (Fig. 2c Knowledge gaps). Species with either scarce or ample information are more likely to be classified as imperiled, supporting the influence of risk aversion in conservation decisions[33]. Previous modeling efforts have described the importance of human knowledge in discriminating between data-sufficient and data-deficient species in the IUCN Red List[34]. In our analysis, the influence of information gaps may be even more relevant as we excluded data-deficient species and examined the role of missing knowledge in real assessment decisions. These results should be interpreted with caution, given that a high proportion of assessed species are missing numerous traits and environmental attributes (e.g., 48% missing ≥30 attributes). Whereas the criteria of the IUCN evaluation framework are not directly represented in these datasets, assessments across species may be inconsistent, in part due to information biases[35]. Results of the preliminary ordinal forest model support at least some influence of the latter, evidenced by high confusion of sub-classifications within imperiled

**Table 1 | Global source datasets, predictor categories, and variables included in machine-learning random forest models to predict freshwater fish imperilment status**

| Dataset | Categories | Sub-categories | Variables |
|---|---|---|---|
| IUCN Red List[26] | Environmental | Habitat | Range perimeter*; Northern, Southern, and Eastern limits; Permanent and Temporary habitats; Habitat types (lotic, lentic, artificial, other); Count of sympatric species*; Count of introduced species*; Count of extinct species |
| FishBase[39,67] | Species$^{T,P,L}$, Socioeconomic$^{V,K}$ | Taxonomic$^T$, Physiology$^P$, Life-history$^L$, Value$^V$, Knowledge$^K$ | Order$^T$; pH minimum (min.), maximum (max.), and range$^P$; water hardness min., max., and range$^P$; body form$^P$; water salinity$^P$; Depth min., max., and range$^P$; Demersal-Pelagic$^L$; Diadromy$^L$; Longevity$^L$; Weight$^P$; Commercial importance$^V$; Aquaculture use$^V$; Bait use$^V$; Aquarium trade$^V$; Game fish$^V$; Electrogenic$^P$; Dangerous$^L$; Feeding type$^L$; Trophic level$^L$; Thermal range$^P$; Locomotion$^P$; Substrate type$^P$ |
| Global Biodiversity Information Facility (GBIF)[40] | Environmental | Habitat | Min., max., and average elevation of occurrences |
| WorldClim v2.1[43] | Environmental | Climate | Annual Mean Temperature (T°), Mean Diurnal T° Range, Isothermality, T° Seasonality, Max T° of Warmest Month, Min T° of Coldest Month, T° Annual Range, Mean T° of Wettest Quarter, Mean T° of Driest Quarter, Mean T° of Warmest Quarter, Mean T° of Coldest Quarter, Annual Precipitation (Precip.), Precip. of Wettest Month, Precip. of Driest Month, Precip. Seasonality, Precip. of Wettest Quarter, Precip. of Driest Quarter, Precip. of Warmest Quarter, Precip. of Coldest Quarter |
| Global River Classification (GloRiC)[44] | Environmental | Hydrology | Count* and mode of geo-hydrological classes, sub-classes and reach types, Climate Moisture Index, Count of lake and wetland reaches*, Max. long-term avg. discharge, Mean flow regime variability, Total stream power |
| Copernicus Global Land Service[68] | Environmental | Habitat | Percent of range cover type (Trees, Shrub, Grass, Crops, Snow, Water, Bare) |
| Global Dam Watch (GDW)[47] | Socioeconomic | Impoundments | Number of dams*, mean discharge, catchment areas |
| HydroWASTE[48] | Socioeconomic | Development | Count of wastewater treatment plants, Total population served, Total discharge treated |
| Global Terrestrial Human Footprint[49] | Socioeconomic | Footprint | Mean human footprint index (1993 and 2009) |
| WorldPop[45] | Socioeconomic | Development | Human population density (early and recent) |
| World Bank[46] | Socioeconomic | Economy | Gross Domestic Product (GDP) 2019, GDP period percent change and slope, GDP Per capita (PC), PC period percent change and slope |
| World Database of Protected Areas[50] | Socioeconomic | Conservation | Protected area percent, Count of Ramsar sites |

Superscript letter codes T (taxonomic), P (physiology), L (life history), V (value), and K (knowledge) indicate the sub-categories associated with each broad category and variable; * indicates range-area adjusted variables (see Table S1). When only one sub-category is listed, all corresponding variables were considered as part of the associated sub-category and broad category for that dataset.

and non-imperiled groupings (Table S1). Furthermore, species assessments exhibit biases with respect to geography and taxonomy, with economically developed regions and early described species having a greater representation[35], which is likely reflected in our findings. To address these alternative explanations, future efforts could focus on attempting to disentangle whether high economic productivity results in higher environmental stress, fewer conservation efforts, or whether economic growth alters perceptions of biodiversity loss.

## Next steps

Future models may be further improved by increasing standardization of listing criteria and the inclusion of more species, with a focus on imperiled species and poorly assessed regions. However, a similar analysis on a prior Red List dataset for 2020 (with 5725 instead of the present 10,631 species) revealed comparable patterns, except for the importance of taxonomic order. A comparison of our model findings with species not yet evaluated by IUCN might provide information about taxa and regions that would benefit most from additional effort for classification. Including additional predictor variables might improve model performance, as well as temporally explicit data, higher-resolution environmental models, and trajectories from historical conditions. Future models could be modified to predict classification histories or future status using historical or scenario datasets,

as well as for predicting the conservation status of data-deficient[36] and yet-to-be-assessed species.

In conclusion, our global analysis of the conservation status of freshwater fish species offers insights into the main factors driving their imperilment or non-imperilment. Our findings underscore the critical role of habitat connectivity, taxonomy, water availability, and low-to-moderate human disturbances in predicting non-imperilment across species. Moreover, the higher model error in identifying imperiled species suggests higher idiosyncrasy in imperilment processes than those observed for secure populations. These findings suggest proactive protection strategies may be more efficient and consistent than reactive conservation approaches, emphasizing the gains that targeted and forward-looking conservation initiatives may create by safeguarding global freshwater fish biodiversity.

## Methods

### Sources and categorization of imperilment and predictor variables

The dataset developed for this study involved the compilation and processing of 12 global data sources (Table 1). Predictive data sources were selected based on their offering of information relevant to one or more of three broad categories: Environmental, Socioeconomic, and Intrinsic. Each of the initial 122 candidate predictive variables in our dataset was also classified into one of 13 sub-categories (i.e., Habitat,

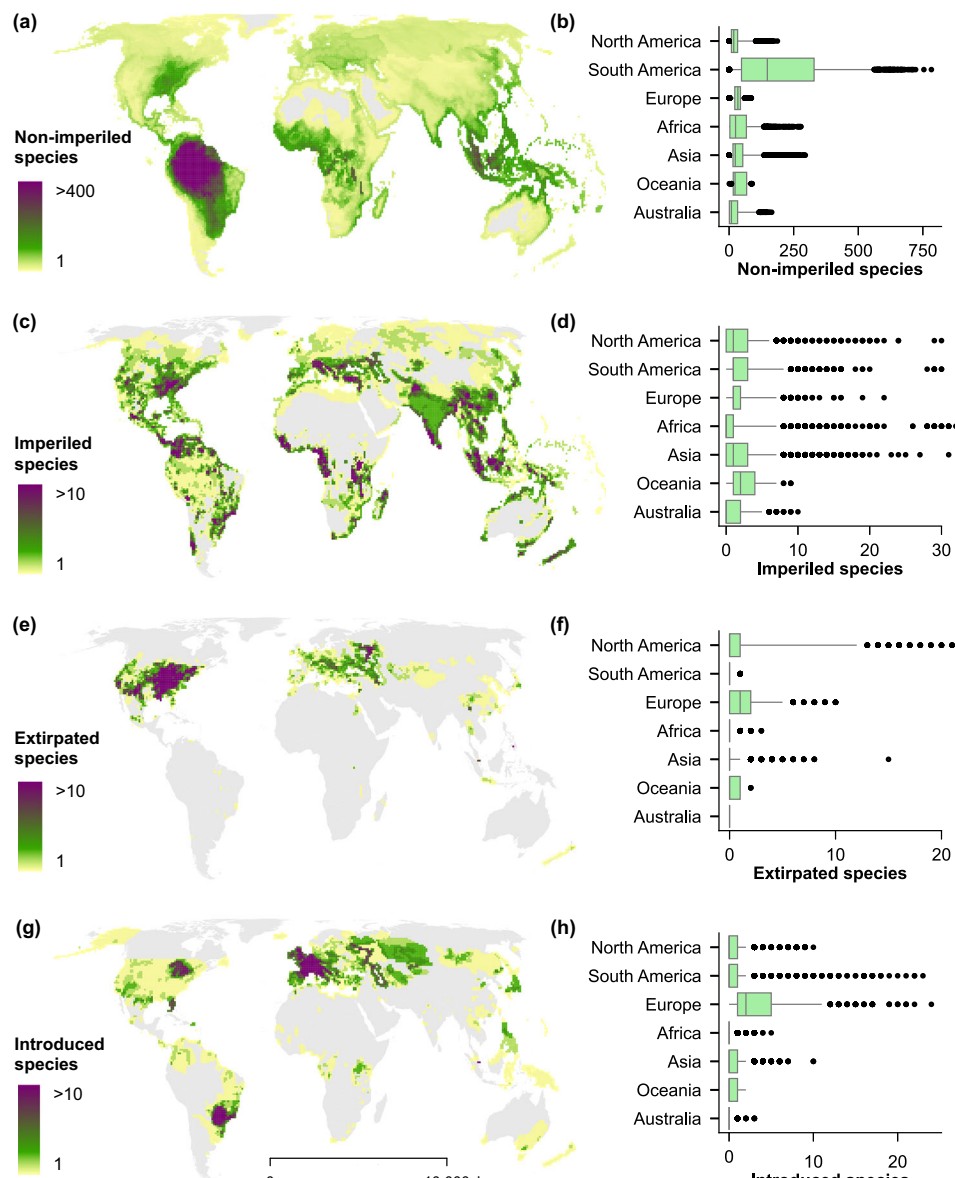

**Fig. 1 | Conservation status of freshwater fishes.** Global distribution of conservation status (panels **a**, **c**, **e**) and introductions of freshwater fishes (panel **g**) in the International Union for Conservation of Nature (IUCN) Red List dataset as of 2024 (v2) at 100-km grid cells[26]. Box-and-whisker plots to the right (panels **b**, **d**, **f**, **h**) show the mean (center line), interquartile ranges (box), 5th and 95th percentiles (whiskers), and outliers (points) in the number of species for cells within each continent. Note that figures only represent assessed species in the spatial dataset of the IUCN Red List, except from data deficient (DD) species.

Climate, Hydrology, Economy, Development, Footprint, Threats, Impoundments, Taxonomy, Physiology, Life-history, Knowledge, Conservation), based on their domain. A predictor representing knowledge gaps was calculated using unknown attributes, the number of predictive variables with NA values, for each species. For the present study, we only used one response variable (i.e., the latest IUCN Red List conservation status), provided as binary (imperiled and non-imperiled) and ordinal (five classes) responses for random forest and ordinal forest models, respectively. For a more detailed description of each variable in the conflated dataset, see Table S1.

**Intrinsic and species response data.** We accessed species conservation data from the IUCN Red List of Threatened Species[26] (2024-v2) using the IUCN Red List API: http://apiv3.iucnredlist.org. Data included the latest conservation assessments, identified threats, and necessary management actions for species recovery. Original threat classes (15) were binned into four larger categories, namely: Human Development,

Exploitation of Natural Resources, Dams and Reservoirs, Invasive Species, and Natural Disasters. Management actions from IUCN were also reclassified into broader classes: Habitat Restoration, Species Control, and Social Policy. Additionally, we used species ranges available through IUCN's Red List spatial database (https://www.iucnredlist.org/resources/spatial-data-download). As a standard requirement, these fish ranges are delineated based on HydroBASINS[37] polygons (basins, sub-basins, and large water bodies[38]). The original dataset of fish ranges was filtered based on uncertainty and origin and combined to represent one merged polygon representing the present, native range of each species ($n = 14,666$). This species list was then filtered to match the list of species available through FishBase ($n = 14,128$). We used the rfishbase package[39] to obtain the species, stock, ecology, and swimming information for each freshwater fish species in the IUCN database based on scientific name (May 20, 2025; v04/2025), removing duplicate entries. We then calculated ranges for values that included minimum and maximum values (e.g., pH, dH).

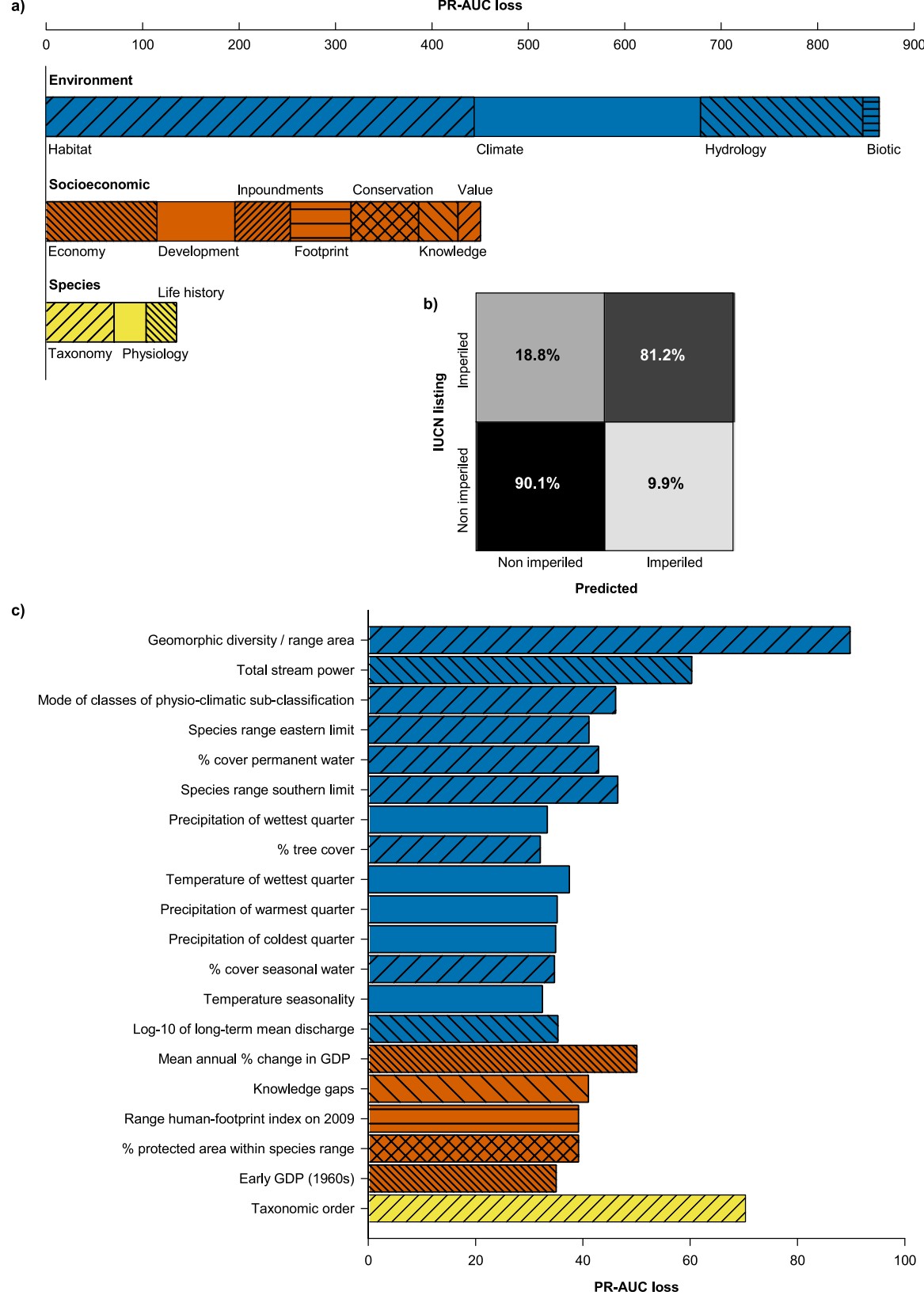

**Fig. 2 | Summary results and variable importance. a** Contributions of predictor categories to accuracy of imperilment assignment (random forest models). PR-AUC is the area under the precision-recall curve. **b** Confusion matrix of global accuracy (88%), misclassification was asymmetric as imperiled species were misclassified more frequently than non-imperiled species. **c** Accuracy contributions from the 20 most important individual predictors colored by category. IUCN refers to the International Union for Conservation of Nature, and GDP to Gross Domestic Product. The color and hash marks for the specific predictors (**c**) correspond to the broad categories (color: Environment, blue; Socioeconomic, orange; and Species, yellow) and sub-categories (hash marks, as labelled) within (**a**).

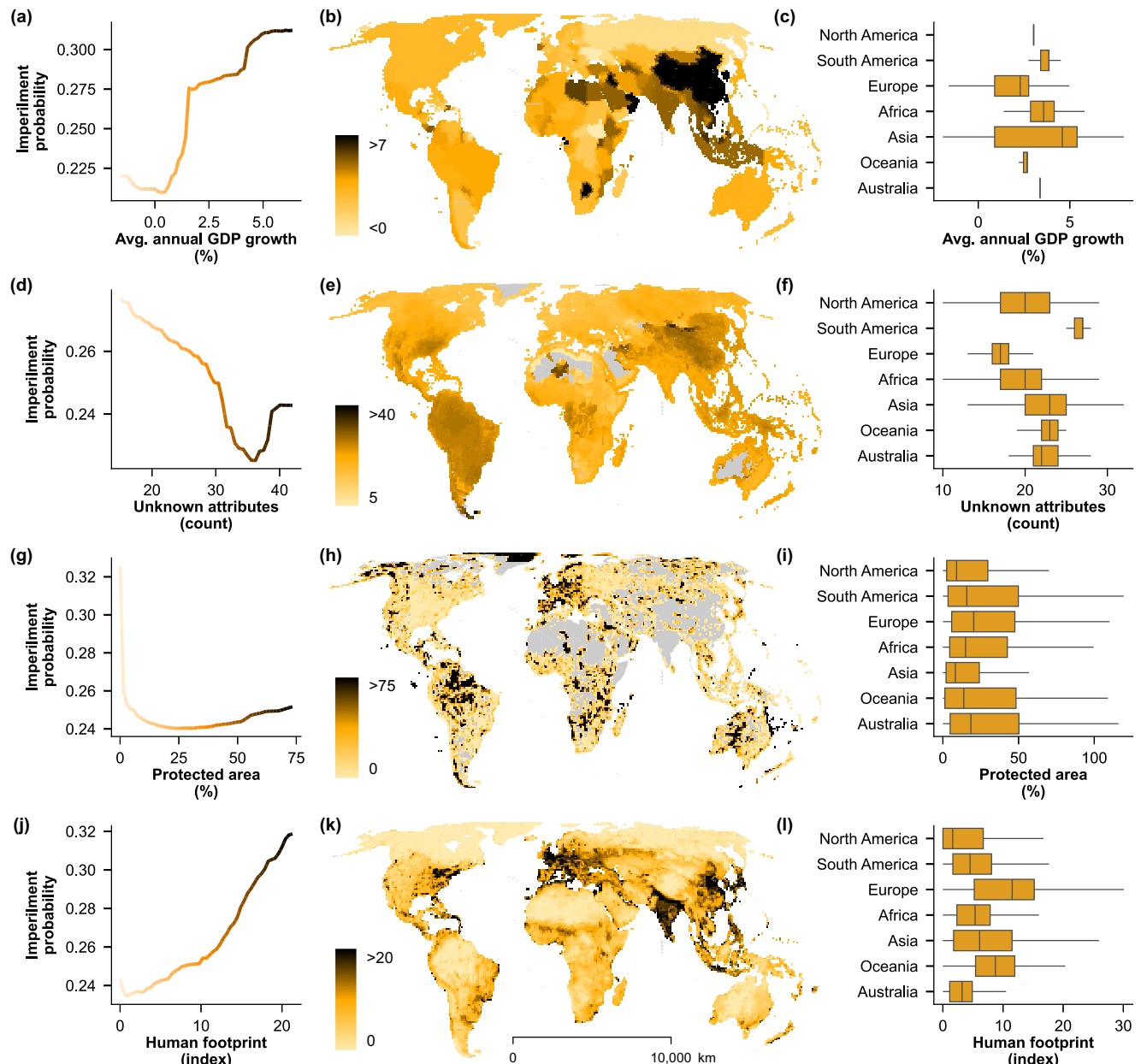

**Fig. 3 | Partial dependence on socioeconomic predictors.** Partial dependence plots (panels **a**, **d**, **g**, **j**) show relationships between the predicted probability of imperilment and the four most important socioeconomic variables, colored on the scale in maps (panels **b**, **e**, **k**) with the spatial distribution of each variable by grid cell. Box-and-whisker plots (panels **c**, **f**, **i**, **l**) show the mean (center line), inter-quartile ranges (box), and 1.5x interquartile ranges (whiskers) for cells within each continent (outliers not shown). Grey represents mapped areas with no value for the mapped variable (panels **e**, **h**; e.g., no protected areas or no species data, due to lack of range). Unknown attributes (panel **d**) are a sum of the NA values for non-spatial predictors from FishBase v4/2024[39,67] and IUCN RedList 2024 v2[26] for a given species (representative of knowledge gaps). Socioeconomic source data (panels **b**, **h**, **k**) derived from publicly available datasets, World Bank, Global Terrestrial Human Footprint, and the World Database of Protected Areas (WDPA)[46,49,50].

To complement the spatial information provided in species ranges, we accessed occurrence locations available through the Global Biodiversity Information Facility (GBIF)[40] via the *rgbif* package in R (as of May 20, 2025). To ensure the precision and quality of spatial coordinates, we queried records ranging from 1990 to 2024 and thoroughly cleansed based on all filters available in the package CoordinateCleaner[41], dropping duplicates, and all records coinciding with country capitals, centroids, scientific collections, as well as coordinates falling outside each species range. This resulted in 407,627 clean occurrences for 8943 species, then used to calculate elevation values (min., max., and mean) using the package elevatr[42] to access AWS (Amazon Web Services) Open Data Terrain Tiles at 100-m pixel size.

**Environmental data.** Non-human environmental data were retrieved from various public repositories, including datasets hosted by Copernicus, WorldClim, and HydroSHEDS. For the characterization of fish ranges based on climatic data, we used WorldClim v2.1[43], a pixel-based bioclimatic model providing 19 variables representing trends in ecologically meaningful climatic conditions, which is often used in species distribution modeling. Datasets for the period 1970–2000 were used as representations of contemporary climatic conditions. For each species, we obtained hydro-geomorphic habitat metrics through the Global River Classification dataset (GloRiC)[44], offering an elevation-derived global hydrography dataset populated with various value-added attributes. Fish ranges were used to summarize GloRiC's reach

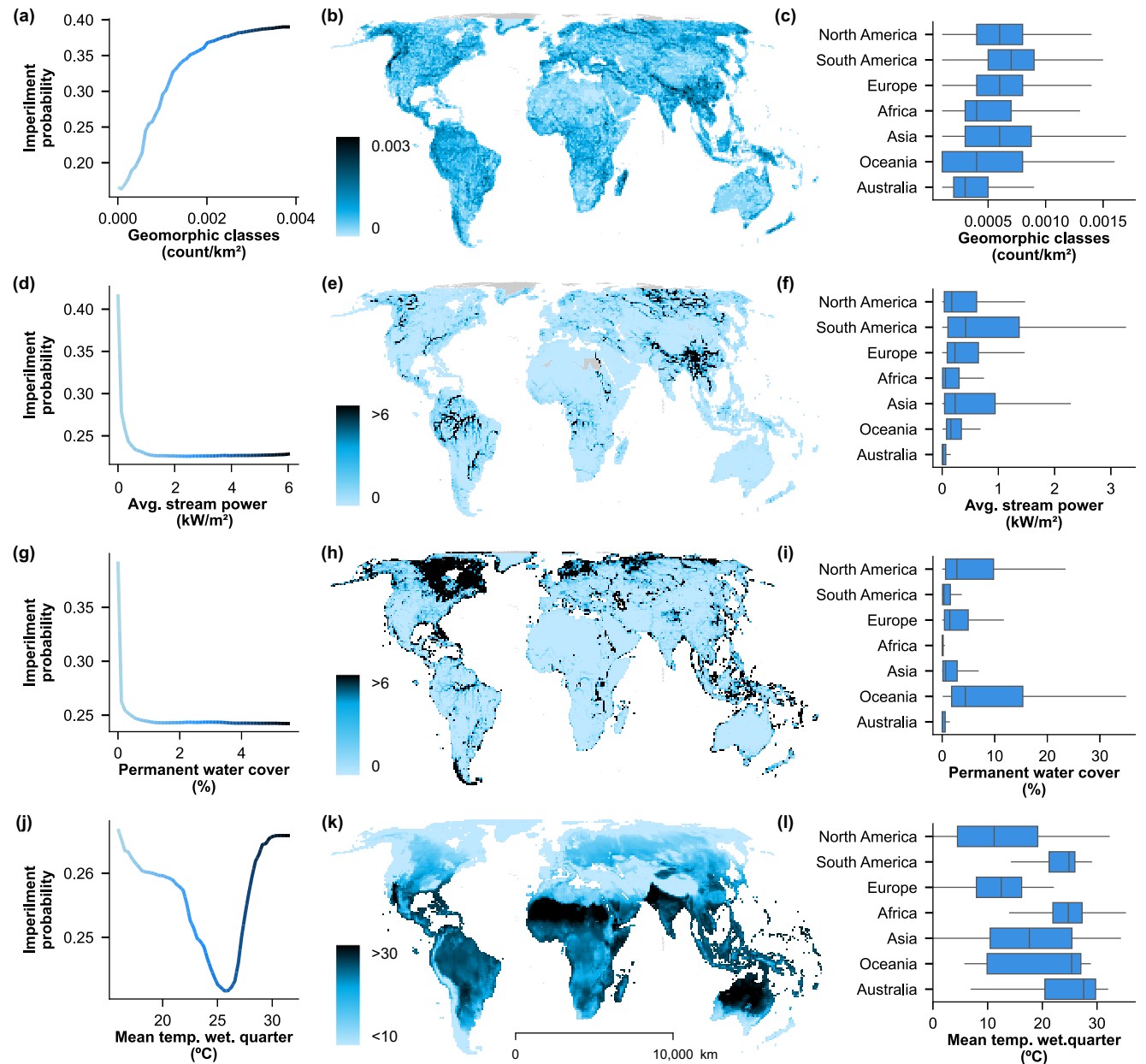

**Fig. 4 | Partial Dependence on Environmental Predictors.** Partial dependence plots (panels **a**, **d**, **g**, **j**) of predicted imperilment probability and four of the most important environmental variables, colored on the scale in maps (panels **b**, **e**, **h**, **k**), with the distribution of variables by grid cell. Box-and-whisker plots (panels **c**, **f**, **j**, **l**) show the mean (center line), interquartile ranges (box), and 1.5x interquartile ranges (whiskers) for cells within each continent (outliers not shown). Value ranges differ between partial dependence plots and map color scales because of the spatial scale of measurement represented (i.e., range-based in partial dependence plots vs. grid-based maps). Mean temp. wet. quarter = mean temperature of the wettest quarter; Avg = average. Environmental source data derived from publicly available datasets, WorldClim v2.1, Global River Classification (GloRiC), and Copernicus Global Land Service[43,44,68].

data, such as counts of unique hydrologic, climatic, and geomorphic reach types as measures of hydro-geomorphic heterogeneity, and mean estimated annual discharge. We used 100-m resolution land cover data from Copernicus Global Land Service (Collection 3, Epoch 2015), derived from PROBA-V satellite observations. We quantified the mean fraction of fish ranges covered by trees, shrubs, grass, crops, bare, snow, and water cover classes.

**Socioeconomic data.** The spatial interaction between species ranges and anthropic variables was assessed based on six main datasets: WorldPop[45], WorldBank[46], the Global Dam Watch[47], HydroWaste[48], Global Terrestrial Human Footprint[49], and the World Database of Protected Areas[50]. WorldPop develops publicly available datasets of

spatial demography, including several indicators of human population dynamics and development indicators. For each species range, we averaged WorldPop's yearly population density values for years 2000 and 2020[45]. Gross Domestic Product (GDP) and GDP per capita as current US dollars were obtained from DataBank, the public database and analysis tool made available by The World Bank[46]. These yearly time series data were assigned to spatially explicit country polygons, as offered by the high-resolution ESRI's Online Service Feature Layer World Countries[51]. We collected human footprint information for each species based on the Global Terrestrial Human Footprint map made available by Venter et al[49] for the years 1993 and 2009. This dataset provides a single index value compositing various measures of human disturbance over the landscape,

including built infrastructure, population density, and agricultural areas, among others. We accessed the Global Dam Watch (GDW) database, a curated source of spatial data on dams and reservoirs[47]. We merged three different datasets for contemporary global data on small dams and reservoirs: the Global geOreferenced Database of Dams (GOODD)[47], Global River Obstruction Database (GROD)[52], and the Joint Research Centre Data (JRC)[53]. GOODD, GROD and JRC contain additional records for smaller dams and reservoirs not captured in other GDW datasets, including all impoundments identifiable on Google Earth imagery, totaling 33,495 data points[47,52,53]. We also used the Global Reservoir and DAm Dataset (GRAND)[54], a curated product of a global collaborative effort mapping existing dams higher than 15 meters and reservoirs bigger than 0.1 cubic kilometers. Its updated version, published in 2019, contains coordinates and accompanying data for 7,424 dams.

We evaluated interactions between fish ranges and wastewater treatment plants using HydroWASTE data[48]. HydroWASTE provides the location and characteristics of 58,502 wastewater treatment plants around the globe, ensuring spatial consistency with the HydroSHEDS network used in this study. We quantified the number of treatment plants within each species range, and two additional metrics: the sum of population served and the sum of river discharge treated.

To quantify the prevalence of protected areas within species ranges, we accessed the World Database on Protected Areas[50] as the most comprehensive and updated source of global protected areas for conservation. Two conservation-related variables were calculated for each species: protected percent and a count of UNESCO's Ramsar Sites within each species range.

## Characterization of fish ranges

The environmental and socioeconomic characterization of fish ranges involved several steps and was scripted in Python 3.11[55] using different packages depending on data formats. For pre-processing of IUCN ranges and most vector-based calculations, we used GIS software (ArcGIS Pro 3.4[51]) functionalities via the arcpy library, while raster-based data sources were processed using the exactextract package[56] to help speed up computations. To focus on freshwater ecosystems, we removed marine regions for marine and estuarine-associated fishes. Fish ranges were then used to summarize underlying spatial data by averaging, adding, counting, or getting ranges, depending on variable types (e.g., continuous, discrete, categorical). All raster data, including climatic variables, land cover classes, and human footprint indexes, were summarized based on the overlapping pixels with fish ranges.

The spatial interaction of fish ranges with dams and reservoirs was captured by calculating the total number of dams, impounded drainage areas, average flows (log), power generation capacity, and reservoir surface areas within the range boundaries of each species. Since most of IUCN's fish ranges consist of large watersheds, calculating the elevation of species' habitats using them would likely give deceiving results, since fish many times use only a few environments within those watersheds. To address this impediment, we used elevation ranges (mean, max., min.) obtained from GBIF occurrence locations, as described above. Lastly, to convert country-level socioeconomic metrics to a range scale, we used a weighted average of the proportional overlap between countries and fish ranges.

## Machine-learning models

Random and ordinal forest algorithms are commonly used in ecology because of their advantages, particularly in dealing with non-linear relationships and interactions[57]. Previous modeling studies have demonstrated their superior performance in classification tasks involving large datasets, including global-scale predictions of IUCN Red List status[36,58,59].

Our initial approach using ordinal forests was implemented using the cforest algorithm in the party package[60]; however, high misclassification rates among adjacent conservation classes (e.g., Least Concern and Vulnerable; See Supplementary Information Table S2) suggested a higher predictability of two response groups, that we refer to here as imperiled (Critically Endangered, Endangered, and Vulnerable) and non-imperiled (Near Threatened and of Least Concern). Respectively, we constructed a binary random forest model, calling the randomForest package[61] through caret[62] for repeated cross-validation and hyperparameter tuning.

**Variable selection and standardization.** The development of random forests was preceded by a variable selection process based on biological relevance, interpretability, and correlation of predictors and response variables. Variables that were expected to contribute directly to the assignment of imperilment (i.e., range area; perimeter) or that contained comparable information (i.e., CITES code; historical listing status; population trend; vulnerability) were removed. We also removed categorical variables with too many levels (i.e., higher than 53) for random forest (i.e., genus, subfamily and family), and those shared by nearly all species (i.e., class).

In order to reduce the influence of species range area in our numerical analysis, variables expected to correspond to range area (i.e., number of sympatric species, introduced species, dams, area of lakes, blocked discharge, dam catchment area, and reservoir surface area, population relying on wastewater treatment plants, and waste discharge) were standardized by dividing over range area, and the correlations of the uncorrected and corrected variables were then compared to range area. The version with the lowest correlation was retained. This reduced all correlations with the range area to less than 30%.

Finally, other highly correlated variables ($|r| > 0.75$) were identified and filtered to remove redundancy and retain the least correlated variables to the rest of the dataset: latest and earliest GDP were removed, GDP slope was retained; latest and earliest GDP per capita were removed, GDP per capita slope was retained; east and west limits were 95% correlated, east limit was retained; count of geomorphic classes and sum of stream network in km were 98% correlated, count of geomorphic classes was retained; a number of historical WorldClim variables were highly correlated (85–98%), mean diurnal temperature range, maximum temperature of the warmest month, mean temperature of the warmest quarter, mean temperature of the coldest quarter, annual temperature range, annual precipitation, precipitation seasonality were removed; mode of classes of physio-climatic sub-classification was 93% correlated with minimum temperature, minimum temperature was retained; human footprint indices for 1993 and 2009 were 97% correlated, human footprint for 2009 was retained; the earliest and latest population densities were 88% correlated, the most recent population density was retained; wastewater population served over range area and wastewater discharge over range area were 89% correlated; wastewater discharge over range area was retained.

**Model construction and weighting.** Random forest models were configured and tuned using the recursive partitioning package caret in R[62]. The random forest model was weighted 3:1 (imperiled:non-imperiled) to account for the greater representation of non-imperiled species and na.roughfix from the randomForest package[61] was used to compensate for gaps in data. For numeric variables, this replaces missing data (NAs) with column medians, while for factor variables, it replaces NAs with the most frequent levels observed, breaking ties at random. We constructed the random forest model with 1500 trees, as we expected more trees to improve performance and were not appreciably limited by run time[63]. Each tree used 63.2% of the data, with the remaining 36.8% of the data used to assess predictive

performance based on various metrics and set to prioritize macro-averaged mean absolute classification errors as a class-unbiased metric for model selection in ordinal regression[64], and the area under the precision-recall curve (PR-AUC) as an unbiased metric for the performance of binary random forest models[65].

Partial dependence tables and plots were generated for all input predictors using the pdp package[66] in R and scaled to assignment probability (e.g., a value of 0.5 corresponded to an even chance of assignment to either category, 0.9 corresponded to a 90% probability of assignment to imperiled). All tuning performance metrics, final model outputs, and partial dependence plots are available in Tables S3–S5.

## Reporting summary

Further information on research design is available in the Nature Portfolio Reporting Summary linked to this article.

## Data availability

The data used in this study were sourced from publicly available datasets: IUCN Red List v2024.2[26], FishBase v4/2024[39,67], Global Biodiversity Information Facility[40], WorldClim[43], Global River Classification[44], Copernicus Global Land Service[68], Global Dam Watch[47], HydroWASTE[48], Global Terrestrial Human Footprint[49], WorldPop[45], WorldBank[46], and the World Database of Protected Areas[50]. No additional datasets were generated in this study. Data questions can be addressed to J. Andres Olivos (andres.olivos@oregonstate.edu).

## Code availability

All processing scripts and output classification models are publicly available at https://github.com/AndresOlivos/freshwater_fish_imperilment_classification and are archived at https://doi.org/10.5281/zenodo.17674411.

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

## Acknowledgements

IA was supported, in part, by the Oregon Agricultural Experiment Station with funding from the Hatch Act capacity funding program, award numbers NI25HFPXXXXXG022 and/or NI25HMFPXXXXXG029, from the USDA National Institute of Food and Agriculture. In-kind support was provided by the US Geological Survey, Maine Cooperative Fish and Wildlife Research Unit. Any use of trade, firm, or product names is for descriptive purposes only and does not imply endorsement by the U.S. Government.

## Author contributions

C.A.M.: conceptualization, methodology, formal analysis, data curation, writing—original draft. J.A.O.: conceptualization, methodology, formal analysis, data curation, writing – original draft, visualization. I.A.: conceptualization, methodology, writing—review & editing, visualization. E.G.B.: conceptualization, methodology, writing—review & editing. S.L.J.: conceptualization, methodology, writing—review & editing. J.D.: conceptualization, methodology, writing—review & editing.

## Competing interests

The authors declare no competing interests.
