## [Transparent Peer Review file · Nature Communications]

Environment, taxonomy, and socioeconomics predict non-imperilment in freshwater fishes

Corresponding Author: Dr Christina Murphy

Version 0:

Reviewer comments:

Reviewer #1

(Remarks to the Author)

The manuscript presents an ambitious attempt to predict the conservation status of over 5,700 freshwater fish species globally, using machine learning and a comprehensive range of environmental, socioeconomic, and intrinsic species-level predictors. While this study provides valuable insights, several areas require substantial improvement.

1. Classification of 'Vulnerable' Species:

The decision to group 'Vulnerable' (VU) species under the 'non-imperiled' category is questionable. According to the IUCN Red List ([https://www.iucnredlist.org/about/faqs#What are the Red List Categories and Criteria](https://www.iucnredlist.org/about/faqs#What%20are%20the%20Red%20List%20Categories%20and%20Criteria)), 'Vulnerable' species are categorized as 'threatened,' along with Endangered (EN) and Critically Endangered (CR) species. The original ordinal model showed that 'Vulnerable' species are more likely to be misclassified into non-imperiled categories. However, this does not justify classifying them as 'non-imperiled.' Please reconsider whether this classification is rational.

2. Species Range and Dataset Discrepancies:

The methodology for merging species ranges and reconciling discrepancies between different datasets, such as GBIF and IUCN, is unclear. After obtaining the range size data from IUCN, were the species ranges merged and cut into grid cells on the map (Figure 1)? Please check the size of the grid cell (10,000 km²?) .

3. Data Cleaning and Merging Process:

Has the GBIF records data been checked and cleaned before merging it with the IUCN data? There is a possibility that some records are from museum collections or have been mis-recorded. The process for merging GBIF data with IUCN range data is also unclear and needs further explanation. Please provide a more detailed description of this process.

4. Figures 3 and 4:

The maps in Figures 3 and 4 were generated from other sources and do not provide sufficient informative data to support the results. I recommend removing these figures and instead merging the partial dependence plots to better convey the findings. The label 'unknown attributes' in Figure 3 is not explained anywhere in the text. Please clarify what these attributes represent.

In Figure 4, the curves in panels a and d are coincidentally identical. Please check the input data for consistency and accuracy.

5. Code and Data Availability:

I strongly recommend that the authors make the code and data available in an online repository, such as GitHub, to ensure reproducibility. This would allow other researchers to access, validate, and build upon the analysis, promoting transparency and fostering future research.

Reviewer #2

(Remarks to the Author)

What are the noteworthy results?

The core idea of using machine learning for predicting species conservation status is promising. However, the results are not clearly presented, and the novelty of the findings is unclear.

Will the work be of significance to the field and related fields? How does it compare to the established literature?

The work could be significant if the methodology were more robust and well-validated. However, it lacks proper comparison to the existing literature and doesn't demonstrate its novelty relative to other approaches.

Does the work support the conclusions and claims, or is additional evidence needed?

The conclusions about the consistency of non-imperilment conditions are not well-supported by evidence. Additional analysis and data (i.e. Validation on independent data set) are needed to back up these claims.

Are there any flaws in the data analysis, interpretation, and conclusions? Do these prohibit publication or require revision?

There are several flaws in the analysis, such as the handling of class imbalance, variable selection, and the model validation process. These flaws require revisions, but they do not necessarily prohibit publication.

Is the methodology sound? Does the work meet the expected standards in your field?

The methodology is not fully sound, as it lacks transparency in key areas such as variable selection, model validation, and comparison to existing methods. In particular, the absence of a clear validation strategy for the model undermines its reliability. Revisions are needed to address these issues and meet the expected standards.

Is there enough detail provided in the methods for the work to be reproduced?

No, the methodology lacks sufficient detail, especially in terms of model specifics and validation. More thorough descriptions are needed to ensure the work can be reproduced.

General Comments

While I believe the core idea of the paper is promising and holds significant potential, I have serious concerns regarding its lack of grounding in the state of the art and, most critically, the methodological approach. The concept of machine learning in the article appears to be introduced without proper integration into the context and without adequate methodological justification. It gives the impression that machine learning is presented as a technological panacea, without careful consideration of its application and without a clear connection to the rest of the research. In general, the methodology, based on Random Forest, overlooks several critical factors as discussed in the following. A major effort is needed to present the statistical aspects of the methodology in a more thorough and clear manner. Many unclear points need to be addressed convincingly to ensure the robustness of the results. In particular, the main concerns revolve around the lack of a result validation strategy, the superficial discussion of the data imbalance, and a variable selection strategy that seems to disregard some fundamental statistical rules. The lack of attention to these details undermines the overall credibility of the approach and consequently of the obtained results. Moreover, there seems to be no discussion of the state of the art or comparison with previous studies, using machine learning or other statistical approaches, which is a fundamental prerequisite for contextualizing the work and demonstrating its novelty and relevance within the field. I believe that, given the significant potential of the dataset and the study's concept, the authors should be afforded an opportunity to address the concerns raised and further strengthen their work. However, it is my assessment that substantial improvements would be required for the manuscript to meet the journal's rigorous standards.

Comments to the Abstract

In general, the abstract would benefit from a more effective inclusion of the practical implications of this research.

Emphasizing these aspects could help demonstrate the broader impact of their findings. What is the most important discovery made in this study? What is the most urgent finding to communicate, and how does it represent a significant advancement in knowledge compared to the current state of the art?

In the statement "we developed a machine-learning model to predict the conservation status of freshwater fishes globally," it would be beneficial to be more specific. Was the model a classifier? Are you adopting a Random Forest-based methodology? Also, since the study focuses on the distinction between imperilment vs. non-imperilment, it would be helpful to specify this early. Declaring upfront that the focus is on the classification of two classes (imperilment vs. non-imperilment) using Random Forest would be optimal.

Comments to the Main

The main section does not provide a detailed context of previous studies, existing models, or alternative approaches to predicting species conservation status. This makes it difficult to position the work within the existing literature and demonstrate how the proposed model adds value.

The term "machine learning" is very broad, yet the authors have been surprisingly vague in presenting their methodology. While a machine learning model is mentioned, there are no details on its nature (e.g., algorithms used, statistical strategy employed, validation methods, performance metrics), and one has to go much further into the paper to find any information. Even then, there is no discussion of the state-of-the-art regarding the application of machine learning or other statistical strategies to the proposed problem. It remains unclear what the model brings that is new, how robust it is, why it is necessary, and how it could be applied in future conservation efforts. The limitations of the model are not addressed. Moreover, non-linear relationships between predictors and conservation status are mentioned but not analyzed in detail. This is critical because a better understanding of such relationships could provide valuable insights.

In general, this section is overly descriptive and lacks analytical depth. For example, significant emphasis is placed on describing global threats (which are already well-documented in the literature), but little attention is given to how the model addresses these threats.

The authors wrote: "The model had a relatively accurate prediction of conservation status of freshwater fishes (85% classification accuracy). However, the prediction of imperilment was far less accurate (67%) than non-imperilment (94%) based on misclassification rates (mis-assignments of test data using the trained model). This mismatch suggests that conditions associated with the non-imperilment of freshwater fishes are more consistent than those driving imperilment. This is a sensible conclusion if fish imperilment is partly represented by species with specific or narrow habitat requirements, consumptive values to humans, or other unique traits contributing to their conservation status." This statement draws a strong conclusion about the consistency of non-imperilment conditions without sufficient evidence or detailed analysis. I

don't understand how to interpret the concept of consistency in this context. In fact, the lower accuracy for imperiled species predictions (67%) may reflect incomplete data, insufficiently representative predictors, data imbalance (Is there any data imbalance? Only in Section 3.2 the authors mention the greater representation of non-imperiled species!), model design limitations, or the inherent complexity of the factors driving imperilment. The model's inability to consistently predict imperiled species suggests that the factors driving imperilment are more intricate and potentially less well-represented in the available data, highlighting the need for more comprehensive data and refined predictors and statistical methodology to improve accuracy. However, no discussion is provided on these important aspects.

Although the authors mention the importance of data gaps and biases, the paper does not explore in depth how these factors directly influence the classifications. In particular, the influence of informational gaps in determining whether a species is classified as imperiled or not is an aspect that warrants more specific investigation. This cannot be postponed to future studies; an analysis of the data gap on the model's performance should be included.

Moreover, the authors have chosen to use a confusion matrix to assess model performance. While the confusion matrix is a widely used tool, there are other advanced statistical metrics available in the literature that might provide more nuanced insights into model performance, such as Precision-Recall AUC, F1-score, which could offer additional information on the model's ability to correctly classify imperiled versus non-imperiled species. It would be valuable to understand why the authors opted for the confusion matrix.

Comments on Sources and categorization of imperilment and predictor variables

The dataset's potential is evident, but a more detailed descriptive analysis is lacking. Statistical summaries and visualizations would enhance the understanding of the dataset's properties.

Comments on Section "Variable Selection and Standardization"

The variable selection process described in the paper raises significant concerns regarding the validity of the proposed methodology. In general, it is well known that the selection of variables based on correlations among the data could introduce bias if the test and/or validation data were also used in the variable selection process. This approach undermines the model's ability to generalize to new data, as it may lead to overfitting and an inaccurate assessment of the model's performance. It is crucial that variable selection be conducted without using test/validation data in order to avoid bias and ensure the validity and reliability of the results. What has been said is certainly valid also in a Cross Validation or Out-Of-Bag (OOB) strategy. Unfortunately, the authors do not discuss this point at all, making it impossible to properly evaluate the methodology they propose. Without proper handling of this aspect, the robustness and reliability of the methodology are severely compromised.

Comments on Model Construction and Weighting

The authors have chosen to rely on the OOB estimate of error for model validation, it would be helpful if the authors could clarify why they have opted for this method instead of conducting a cross-validation strategy (e.g., k-fold cross-validation). In fact, cross-validation might provide a more comprehensive assessment of model generalizability, particularly if the dataset exhibits heterogeneity or class imbalance. A discussion of why OOB was preferred over cross-validation, and whether this choice was validated through any comparison with cross-validation performance, would strengthen the methodology and provide more transparency regarding the robustness of the model's results.

Additionally, validating the results on independent data is crucial to assess the robustness and real-world applicability of the model. Unfortunately, it does not appear that the authors have even considered this task in their work. There is no mention of any external validation on independent datasets, nor is there any discussion of potential concerns related to the generalizability of the model beyond the data on which it was trained. This oversight is very critical, as without such validation, the model's ability to perform effectively in real-world conservation efforts remains uncertain. The lack of a validation task, along with the previous concerns, casts several doubts on the reliability of the results and undermines the overall trustworthiness of the methodology presented.

Version 1:

Reviewer comments:

Reviewer #1

(Remarks to the Author)

The authors have done a commendable job in comprehensively addressing all the concerns raised. Congratulations on this excellent work.

Guohuan Su

(Remarks on code availability)

Reviewer #2

(Remarks to the Author)

I thank the authors for their substantial revisions, which have greatly improved the manuscript. The updated version now

presents a transparent and reproducible methodology, including clearer model construction, cross-validation, robust metrics to address class imbalance, and open access to code and data. The study is better framed within the state of the art, and its limitations—particularly regarding prediction of imperiled species and potential data biases—are openly discussed. In my view, the manuscript now reaches a standard suitable for publication and I recommend acceptance after minor editorial revisions.

(Remarks on code availability)

We appreciate the opportunity to resubmit our substantially revised manuscript. In this revision, we updated the species dataset to include the most recent IUCN records (increasing from approximately 5,700 to 10,631 fishes) and reran all analyses accordingly. These updates produced results consistent with our earlier findings, while highlighting the high importance of taxonomy (order) as a predictor. The additional data increased both the robustness of our conclusions and the broader applicability of our modeling framework. We respond to the helpful and specific reviewer comments below.

We refer to line numbers for the clean revised manuscript draft below.

REVIEWER COMMENTS

Reviewer #1 (Remarks to the Author):

The manuscript presents an ambitious attempt to predict the conservation status of over 5,700 freshwater fish species globally, using machine learning and a comprehensive range of environmental, socioeconomic, and intrinsic species-level predictors. While this study provides valuable insights, several areas require substantial improvement.

1. Classification of 'Vulnerable' Species:

The decision to group 'Vulnerable' (VU) species under the 'non-imperiled' category is questionable. According to the IUCN Red List ([https://www.iucnredlist.org/about/faqs#What are the Red List Categories and Criteria](https://www.iucnredlist.org/about/faqs#What%20are%20the%20Red%20List%20Categories%20and%20Criteria)), 'Vulnerable' species are categorized as 'threatened,' along with Endangered (EN) and Critically Endangered (CR) species. The original ordinal model showed that 'Vulnerable' species are more likely to be misclassified into non-imperiled categories. However, this does not justify classifying them as 'non-imperiled.' Please reconsider whether this classification is rational.

Response: We agree with the reviewer's observation and have revised the analysis to include the 'Vulnerable' category within the imperiled class of the binary classification model, in line with IUCN Red List definitions.

This is noted in the Abstract line 30 and Methods line 99, as well as in the Supplementary Information Tables S2 and S3.

To further address this point, we now describe the ordinal forest approach in more detail in Methods and provide full results in the supplementary material for comparison across classification schemes.

See Methods line 456 and Table S2.

2. Species Range and Dataset Discrepancies:

The methodology for merging species ranges and reconciling discrepancies between different datasets, such as GBIF and IUCN, is unclear. After obtaining the range size

data from IUCN, were the species ranges merged and cut into grid cells on the map (Figure 1)? Please check the size of the grid cell (10,000 km²?).

Response: We clarified that IUCN range polygons were processed individually to calculate summary statistics from each geospatial dataset. Figure 1 now explicitly shows counts of overlapping ranges at each grid cell in the maps, roughly 16,400 100km grid cells (10,000 km²). The Figure 1 caption and Methods (lines 357-375) have been updated for clarity.

3. Data Cleaning and Merging Process:

Has the GBIF records data been checked and cleaned before merging it with the IUCN data? There is a possibility that some records are from museum collections or have been mis-recorded. The process for merging GBIF data with IUCN range data is also unclear and needs further explanation. Please provide a more detailed description of this process.

Response: GBIF occurrence records were extensively tested and filtered using the R package '*CoordinateCleaner*' (Zizka et al. 2019). Among these filters, we removed GBIF occurrences falling outside each species' extant, native IUCN range. Details have now been added to the Methods section, lines 371-373.

4. Figures 3 and 4:

The maps in Figures 3 and 4 were generated from other sources and do not provide sufficient informative data to support the results. I recommend removing these figures and instead merging the partial dependence plots to better convey the findings.

Response: Figures 3 and 4 were revised for clearer interpretation and procedural transparency. The maps now summarize the distribution of top environmental predictors sourced from the same datasets used to characterize species ranges. To better convey the relationship between maps and PDPs, we colored dependence functions using the same color scale as the maps.

The label 'unknown attributes' in Figure 3 is not explained anywhere in the text. Please clarify what these attributes represent.

Response: (b) The variable 'unknown attributes' (i.e., number of empty fields for each species) is now described in the methods. See Methods lines 218-219.

In Figure 4, the curves in panels a and d are coincidentally identical. Please check the input data for consistency and accuracy.

Response: (c) The input data for the plots was checked to confirm they were correct. Both variables show similar relationships with imperilment probability, with subtle differences that are not noticeable in the partial dependence plot. See Figure 4 and Table S5.

5. Code and Data Availability:

I strongly recommend that the authors make the code and data available in an online repository, such as GitHub, to ensure reproducibility. This would allow other researchers to access, validate, and build upon the analysis, promoting transparency and fostering future research.

Response: The revised code and associated datasets are available at <https://gitfront.io/r/AndresOlivos/BTK8dxjNenih/freshwater-fish-imperilment-classification/> and will be publicly available through https://github.com/AndresOlivos/freshwater_fish_imperilment_classification upon acceptance.

Reviewer #2 (Remarks to the Author):

What are the noteworthy results?

The core idea of using machine learning for predicting species conservation status is promising. However, the results are not clearly presented, and the novelty of the findings is unclear.

Response: We appreciate the reviewer's constructive criticism and recognition of the potential of the study. We carefully addressed all comments through a revised analysis and have made substantial revisions to strengthen the methodology, clarify novelty, and improve presentation. We are confident the reviewed manuscript clearly conveys the novelty and importance of our findings.

As examples, we have added a significance statement and see revised lines 139-206, in particular 145-152.

Will the work be of significance to the field and related fields? How does it compare to the established literature?

The work could be significant if the methodology were more robust and well-validated. However, it lacks proper comparison to the existing literature and doesn't demonstrate its novelty relative to other approaches.

Response: Study methods were extensively revised based on reviewers' feedback, and the reviewed manuscript presents a comprehensive description of model development and validation, all framed within the relevant literature.

See additions of references 42,43,45,46, and 47 and associated lines 157-166 and 169-186.

Does the work support the conclusions and claims, or is additional evidence needed?

The conclusions about the consistency of non-imperilment conditions are not well-supported by evidence. Additional analysis and data (i.e. Validation on independent data set) are needed to back up these claims.

Response: We strengthened the interpretation of a higher predictability of non-imperilment and are confident the revised manuscript presents more defensible conclusions in this regard. In short, environmental conditions in secure species' ranges tend to be more homogeneous than those of imperiled species. This interpretation is now better supported by the modeling procedure (using cross-validation and performance metrics robust to class imbalance) and a holdout validation test (with 20% of the dataset), as described in the methods. These changes improve both the empirical basis and the interpretive clarity of our conclusions.

See lines 139-141, Figure 2b, Supplementary Information, lines 450-460 and code as detailed in the GitHub repository (<https://gitfront.io/r/AndresOlivos/BTK8dxjNenih/freshwater-fish-imperilment-classification/>).

Are there any flaws in the data analysis, interpretation, and conclusions? Do these prohibit publication or require revision?

There are several flaws in the analysis, such as the handling of class imbalance, variable selection, and the model validation process. These flaws require revisions, but they do not necessarily prohibit publication.

Response: All these methodological concerns were carefully addressed in the revised manuscript. See additions describing the approach to class imbalance, variable selection approach, and model validation.

See section 3 in Methods.

Is the methodology sound? Does the work meet the expected standards in your field?

The methodology is not fully sound, as it lacks transparency in key areas such as variable selection, model validation, and comparison to existing methods. In particular, the absence of a clear validation strategy for the model undermines its reliability. Revisions are needed to address these issues and meet the expected standards.

Response: As replied above, we are confident the revised manuscript now meets transparency and validation standards, while justifying the selected methodology (see text).

Full code is available here: <https://gitfront.io/r/AndresOlivos/BTK8dxjNenih/freshwater-fish-imperilment-classification/>

Is there enough detail provided in the methods for the work to be reproduced?

No, the methodology lacks sufficient detail, especially in terms of model specifics and validation. More thorough descriptions are needed to ensure the work can be reproduced.

Response: We now ensure the reproducibility of our study through a more comprehensive methods' description and make available all the code used for dataset preparation (see above), modeling, and analysis. See revised Methods, Supplementary Material, and full code available here:

<https://gitfront.io/r/AndresOlivos/BTK8dxjNenih/freshwater-fish-imperilment-classification/>

General Comments

While I believe the core idea of the paper is promising and holds significant potential, I have serious concerns regarding its lack of grounding in the state of the art and, most critically, the methodological approach. The concept of machine learning in the article appears to be introduced without proper integration into the context and without adequate methodological justification. It gives the impression that machine learning is presented as a technological panacea, without careful consideration of its application and without a clear connection to the rest of the research. In general, the methodology, based on Random Forest, overlooks several critical factors as discussed in the following. A major effort is needed to present the statistical aspects of the methodology in a more thorough and clear manner. Many unclear points need to be addressed convincingly to ensure the robustness of the results. In particular, the main concerns revolve around the lack of a result validation strategy, the superficial discussion of the data imbalance, and a variable selection strategy that seems to disregard some fundamental statistical rules. The lack of attention to these details undermines the overall credibility of the approach and consequently of the obtained results. Moreover, there seems to be no discussion of the state of the art or comparison with previous studies, using machine learning or other statistical approaches, which is a fundamental prerequisite for contextualizing the work and demonstrating its novelty and relevance within the field. I believe that, given the significant potential of the dataset and the study's concept, the authors should be afforded an opportunity to address the concerns raised and further strengthen their work. However, it is my assessment that substantial improvements would be required for the manuscript to meet the journal's rigorous standards.

Response: We have added additional details in the methods, including more in-depth descriptions of our approach to class imbalance, variable selection, and model validation as well as context. We also now ensure the reproducibility of our study through a more comprehensive methods' description and make available all the code used for dataset preparation, modeling, and analysis as noted.

Full code is available here: <https://gitfront.io/r/AndresOlivos/BTK8dxjNenih/freshwater-fish-imperilment-classification/>

Comments to the Abstract

In general, the abstract would benefit from a more effective inclusion of the practical implications of this research. Emphasizing these aspects could help demonstrate the broader impact of their findings. What is the most important discovery made in this study? What is the most urgent finding to communicate, and how does it represent a significant advancement in knowledge compared to the current state of the art? In the statement "we developed a machine-learning model to predict the conservation status of freshwater fishes globally," it would be beneficial to be more specific. Was the model a classifier? Are you adopting a Random Forest-based methodology? Also, since

the study focuses on the distinction between imperilment vs. non-imperilment, it would be helpful to specify this early. Declaring upfront that the focus is on the classification of two classes (imperilment vs. non-imperilment) using Random Forest would be optimal.

Response: Although word-limited, the abstract now specifies that our approach uses ordinal and binary random forests to classify imperiled and non-imperiled species. Both methods provide consistent stories on the importance of habitat, taxonomic order, water, and disturbance for non-imperilment, but the ordinal forest showed higher confusion in assignment, especially for some imperiled classes. Additional methodological details, including ordinal model results, are provided in the supplementary material.

See Abstract lines 28-34, Supplementary Information Tables S2-S5, and <https://gitfront.io/r/AndresOlivos/BTK8dxjNenih/freshwater-fish-imperilment-classification/>

Comments to the Main

The main section does not provide a detailed context of previous studies, existing models, or alternative approaches to predicting species conservation status. This makes it difficult to position the work within the existing literature and demonstrate how the proposed model adds value.

Response: Though constrained by the word count limit: we were able to expand on the novelty of our findings within the context of relevant global conservation studies (e.g., lines 155-166, 169-175, 177-183), and alternative modeling approaches (e.g., lines 451-453).

Also see added references 42, 43, 45, 46, 47, 60, and 61.

The term "machine learning" is very broad, yet the authors have been surprisingly vague in presenting their methodology. While a machine learning model is mentioned, there are no details on its nature (e.g., algorithms used, statistical strategy employed, validation methods, performance metrics), and one has to go much further into the paper to find any information. Even then, there is no discussion of the state-of-the-art regarding the application of machine learning or other statistical strategies to the proposed problem. It remains unclear what the model brings that is new, how robust it is, why it is necessary, and how it could be applied in future conservation efforts. The limitations of the model are not addressed.

Response: The model types are now specified in the abstract and main text, with comprehensive details in the methods, supplement, and GitHub repository, reflecting all revisions made to the modeling approach.

See revised Abstract, Methods section 3, and lines 80-110.

Moreover, non-linear relationships between predictors and conservation status are mentioned but not analyzed in detail. This is critical because a better understanding of such relationships could provide valuable insights.

Response: We included a more thorough description of partial dependence relationships between the most important predictors and the response variable, as well as a supplementary table with described patterns and interpretations for all variables (Table S5). See Supplementary Information Table S5.

In general, this section is overly descriptive and lacks analytical depth. For example, significant emphasis is placed on describing global threats (which are already well-documented in the literature), but little attention is given to how the model addresses these threats.

Response: The revised manuscript places more emphasis on connecting findings with the relevant literature on species threats. See additions of references 42,43,45,46, and 47 and associated lines 157-166 and 169-186.

The authors wrote: *“The model had a relatively accurate prediction of conservation status of freshwater fishes (85% classification accuracy). However, the prediction of imperilment was far less accurate (67%) than non-imperilment (94%) based on misclassification rates (mis-assignments of test data using the trained model). This mismatch suggests that conditions associated with the non-imperilment of freshwater fishes are more consistent than those driving imperilment. This is a sensible conclusion if fish imperilment is partly represented by species with specific or narrow habitat requirements, consumptive values to humans, or other unique traits contributing to their conservation status.”* This statement draws a strong conclusion about the consistency of non-imperilment conditions without sufficient evidence or detailed analysis. I don't understand how to interpret the concept of consistency in this context. In fact, the lower accuracy for imperiled species predictions (67%) may reflect incomplete data, insufficiently representative predictors, data imbalance (Is there any data imbalance? Only in Section 3.2 the authors mention the greater representation of non-imperiled species!), model design limitations, or the inherent complexity of the factors driving imperilment. The model's inability to consistently predict imperiled species suggests that the factors driving imperilment are more intricate and potentially less well-represented in the available data, highlighting the need for more comprehensive data and refined predictors and statistical methodology to improve accuracy. However, no discussion is provided on these important aspects.

Response: We did account for data unbalance using class weighting and now use robust metrics for parameter tuning and performance assessment (e.g., Precision-Recall AUC, F1). See text additions describing the approach to class imbalance, variable selection approach, and model validation in lines 141-143 and lines 494-510. We also agree with the reviewers' interpretation of model confusion as described in the abstract (lines 32-35) and main text (lines 139-149 and 201-203).

Although the authors mention the importance of data gaps and biases, the paper does not explore in depth how these factors directly influence the classifications. In particular, the influence of informational gaps in determining whether a species is classified as imperiled or not is an aspect that warrants more specific investigation. This cannot be postponed to future studies; an analysis of the data gap on the model's performance should be included.

Response: We have reworded for clarity, as this refers to 'knowledge gaps', a predictor variable to capture the number of missing values across all variables (many not included in the model) for that species. As noted, in many cases, these values correspond to predictors that were not included in the model due to missing data. As such, we can only check for the importance of our variable gauging human knowledge, not the originating data gaps, as they are not in the model. Our statement is intended to capture the importance of perceived 'unknowns', not missing values in the model per se.

See lines 66-69, 167-186, 218-219, and 342-343.

Moreover, the authors have chosen to use a confusion matrix to assess model performance. While the confusion matrix is a widely used tool, there are other advanced statistical metrics available in the literature that might provide more nuanced insights into model performance, such as Precision-Recall AUC, F1-score, which could offer additional information on the model's ability to correctly classify imperiled versus non-imperiled species. It would be valuable to understand why the authors opted for the confusion matrix.

Response: We opted for the confusion matrix in the results figure as a more comprehensive summary of performance by class. However, we did update the metric used for tuning parameters during model training to PR-AUC to better address class imbalance, as noted in the methods, and more detailed metrics can now be found in the supplementary material.

See description of performance metrics in lines 501-505 and results in Tables S2 and S3.

Comments on Sources and categorization of imperilment and predictor variables

The dataset's potential is evident, but a more detailed descriptive analysis is lacking. Statistical summaries and visualizations would enhance the understanding of the dataset's properties.

Response: The code to compile the dataset, construct the models, and the model outputs are now available here (<https://gitfront.io/r/AndresOlivos/BTK8dxjNenih/freshwater-fish-imperilment-classification/>). Additional information on the predictor variables related to the model, including partial dependence plots are available in the supplementary material (Table S5).

Comments on Section "Variable Selection and Standardization"

The variable selection process described in the paper raises significant concerns regarding the validity of the proposed methodology. In general, it is well known that the selection of variables based on correlations among the data could introduce bias if the test and/or validation data were also used in the variable selection process. This approach undermines the model's ability to generalize to new data, as it may lead to overfitting and an inaccurate assessment of the model's performance. It is crucial that variable selection be conducted without using test/validation data in order to avoid bias and ensure the validity and reliability of the results. What has been said is certainly valid also in a Cross Validation or Out-Of-Bag (OOB) strategy. Unfortunately, the authors do not discuss this point at all, making it impossible to properly evaluate the methodology they propose. Without proper handling of this aspect, the robustness and reliability of the methodology are severely compromised.

Response: Doubling the number of species (to over 10,000, including 2,640 imperiled species) by updating to the latest IUCN spatial data release has increased our confidence in a model built and validated by holding out a subset of the dataset. We have designated a random, balanced 80%/20% training-test split to the dataset prior to modeling.

See <https://gitfront.io/r/AndresOlivos/BTK8dxjNenih/freshwater-fish-imperilment-classification/> and the Objectives and Modeling Framework as detailed in the repository.

Comments on Model Construction and Weighting

The authors have chosen to rely on the OOB estimate of error for model validation, it would be helpful if the authors could clarify why they opted for this method instead of conducting a cross-validation strategy (e.g., k-fold cross-validation). In fact, cross-validation might provide a more comprehensive assessment of model generalizability, particularly if the dataset exhibits heterogeneity or class imbalance. A discussion of why OOB was preferred over cross-validation, and whether this choice was validated through any comparison with cross-validation performance, would strengthen the methodology and provide more transparency regarding the robustness of the model's results.

Response: The revised model presented in the current manuscript was developed using repeated cross validation (5 folds × 5 repeats) instead of a single OOB estimate. No major changes were observed in overall model performance between validation procedures.

See <https://gitfront.io/r/AndresOlivos/BTK8dxjNenih/freshwater-fish-imperilment-classification/> and results presented in Tables S2 and S3.

Additionally, validating the results on independent data is crucial to assess the robustness and real-world applicability of the model. Unfortunately, it does not appear that the authors have even considered this task in their work. There is no mention of any external validation on independent datasets, nor is there any discussion of potential concerns related to the generalizability of the model beyond the data on which it was

trained. This oversight is very critical, as without such validation, the model's ability to perform effectively in real-world conservation efforts remains uncertain. The lack of a validation task, along with the previous concerns, casts several doubts on the reliability of the results and undermines the overall trustworthiness of the methodology presented.

Response: We now provide a validation test on independent data by random, balanced, 80% training, 20% test split. Associated code and data are available at (<https://gitfront.io/r/AndresOlivos/BTK8dxjNenih/freshwater-fish-imperilment-classification/>). Again, no major changes were observed in overall model performance or results.